# Multi-Span Question Answering using Span-Image Network

## Abstract

Question-answering (QA) models aim to find an answer given a question and context. Language models like BERT are used to associate question and context to find an answer span. Prior art on QA focuses on finding the best answer. There is a need for multi-span QA models to output the top-K likely answers to questions such as "*Which companies Elon Musk started?*" or "*What factors cause global warming?*" In this work, we introduce Span-Image architecture that can learn to identify multiple answers in a context for a given question. This architecture can incorporate prior information about the span length distribution or valid span patterns (*e.g., end index has to be larger than start index*), thus eliminating the need for post-processing. Span-Image architecture outperforms the state-of-the-art in top-K answer accuracy on SQuAD dataset and in multi-span answer accuracy on an Amazon internal dataset.

## 1 Introduction

Answering questions posted as text to search engines or spoken to virtual assistants like Alexa has become a key feature in information retrieval systems. Publicly available reading comprehension datasets including WikiQA (Yang et al., 2015), TriviaQA (Joshi et al., 2017), NewsQA (Trischler et al., 2016), and SQuAD (Rajpurkar et al., 2016) have fostered research in QA models. SQuAD is one of the most widely-used reading comprehension benchmarks that has an active leaderboard with many participants. Even though there are models that beat human-level accuracy in SQuAD, these QA systems can do well by learning only context and type-matching heuristics (Weissenborn et al., 2017) but may still be far from true language understanding since they do not offer robustness to adversarial sentences (Jia & Liang, 2017). To better measure performance, SQuAD v2.0 Rajpurkar et al. (2018) extends v1.1 by allowing questions that have no explicit answers in a given paragraph.

QA can be modeled as a task to predict the span (i.e., start and end indices) of an answer given a question and an input paragraph. To find the answer span, language representation models such as BERT can be used to associate a question with a given paragraph Devlin et al. (2019). BERT is pre-trained on unsupervised tasks using large corpora. Its input representation permits a pair, which is well suited for having a question and a passage as input. By fine-tuning BERT on SQuAD, a QA model can be obtained. Questions without an answer are treated as having a span that begins and ends with the special BERT token: $[CLS]$. In this way, a BERT-based QA model can offer an actual answer or 'no-answer" to all questions in SQuAD v1.1 and v2.0 datasets.

Prior work on QA assumes presence of a single answer or lack of any answer Seo et al. (2016), Devlin et al. (2019). Furthermore, they assume a separable probability distribution function (*pdf*) for start and end indices of an answer span, which leads to a separable loss function. This approach has two major disadvantages: 1) It prevents the QA model from predicting multiple spans without post-processing. 2) Since a separable *pdf* is used, the QA model can not learn to evaluate compatibility of start and end indices, thus suffering from performance degradation. Pang et al. (2019) consider a hierarchical answer span by sorting the product of start and end probabilities to support multiple spans. However, they still assume a separable *pdf* for start and end indices. To the best of our knowledge, a multi-span QA architecture has not been proposed.

We introduce Span-Image architecture to enable multi-span answers (or multiple answers) given a question and a paragraph. Each pixel (i,j) in the span-image corresponds to a span starting at $i$th position and ending at $j$th. Typical image processing networks like 2D convolutional network layers

are used. Span-Image architecture enables the model to couple start and end indices to check for their compatibility. Constraints such as "*the end index has to be bigger than the start index*", can be automatically embedded into the model. Moreover, other span characteristics such as "*shorter answers are more likely to occur*" (see Figure 1), can be learned by the model, thus eliminating the need for post-processing or regularization.

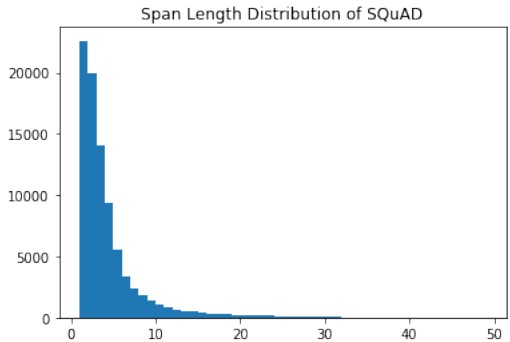

Figure 1: Span length histogram on SQuAD shows that shorter answers are more likely to occur. Span-Image architecture can incorporate this prior information since each output pixel predicts a span whose length is known by the model.

Our contributions are summarized as below:

- We present Span-Image architecture, a novel method that enables multi-span answer prediction.
- Specially designed image channels in the proposed architecture can help the QA model capture span-characteristics and eliminate the need for post-processing.
- Span-Image network is modular and can be added to most DNN-based QA models without requiring changes to the previous layers.

## 2 MULTI-SPAN PREDICTION

In this section we first highlight BERT QA architecture, and present our span-image architecture that consumes BERT outputs.

### 2.1 BERT QA ARCHITECTURE

The QA task in BERT uses a separable *pdf*: $p(s^S, s^E) = p(s^S) \times p(s^E)$ where $s^S$ and $s^E$ denotes one-hot variables of length $N$ for start and end indices for a paragraph of length $N$, respectively. Therefore, BERT QA architecture assumes start and end index probabilities to be independent from each other. Given predicted probabilities $p_{BERT}(s^S)$ and $p_{BERT}(s^E)$ as outputs of BERT, a question $q$ of length $M$ and a passage $g$ of length $N$, the QA loss fuction for fine-tuning BERT is then given by

$$Loss(q, p, t^S, t^E) = H(t^S, p_{BERT}(s^S)) + H(t^E, p_{BERT}(s^E)), \tag{1}$$

where $H$ is the cross-entropy function, and $t$ is target span with start and end indices $t^S$ and $t^E$, respectively.

BERT has two separate outputs for start and end indices, which makes it impossible for the model to check for compatibility of $s^S$ and $s^{E}$[1] or utilize information such as span length (*i.e.*, $s^E - s^S$) in its predictions. Figure 2 shows BERT QA architecture.

---

[1] One can claim attention heads in the transformer network will correlate tokens but this does not happen explicitly as in our architecture, where probability for each possible span is computed jointly.

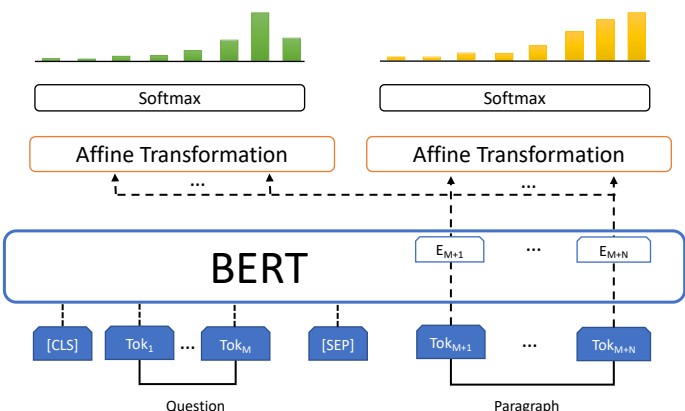

Figure 2: BERT QA architecture predicts start and end index probability distributions separately.

## 2.2 SPAN-IMAGE NETWORK

Span-image network does not dictate a separable *pdf* for start and end indices, hence $p(s^S, s^E) \neq p(s^S) \times p(s^E)$. Given a question $q$ and paragraph $g$, BERT outputs $D$ dimensional vector sequence $BERT(q, g)$ of length $M + N + 2$ (see Figure 2). Let's denote the last $N$ vectors in the sequence, which corresponds to paragraph $g$, with $BERT^g(q, g)$. Using two affine transformations denoted by $W^S$ and $W^E$, each of which has $D$ units, we create 2 vector sequences $W^S(BERT^g(q, g))$ and $W^E(BERT^g(q, g))$ of length $N$. A pixel at location $(i, j)$ has $D$ channels and is given by

$$span\_im_{i,j} = W^S(BERT^g(q, g))_i \circ W^E(BERT^g(q, g))_j, \tag{2}$$

where $\circ$ denotes element-wise multiplication of $D$-dimensional vectors in $i^{th}$ and $j^{th}$ locations of $W^S(BERT^g(q, g))$ and $W^E(BERT^g(q, g))$, respectively. Hence, span-image $span\_im$ shown in Figure 3, is a 3-dimensional tensor of depth $D$ and of height and width $N$. This enables us to borrow techniques such as 2-dimensional convolutional filtering, max-pooling, and ReLU from convolutional neural network (CNN) architectures for image classification. The output of the span-image network is an $N \times N$ logit-image, $logit\_im$, with a single channel (*i.e.,* a logit for each possible pixel/span).

Each channel in $span\_im$ is a matrix of rank 1. Therefore, each channel is separable and has limited potential beyond the separable approach described in Section 2.1. However, applying two-dimensional convolutional layers improves performance and makes $logit\_im$ non-separable, thus eliminating the independence assumption on start and end indices.

The probability of each span can be computed by applying sigmoid function on each pixel or softmax in $logit\_im$. Using sigmoid makes no assumption on number of spans, while using softmax assumes a single span in every paragraph. The best function to use depends on the QA dataset. For example, in our experiments, using softmax gave us best results for fine-tuning BERT on SQuAD while sigmoid performed better on our internal multi-span datasets. Denoting $p(s^S = i, js^E = j)$ by $p_{i,j}$ for simplicity, span probabilities for single-span and multi-span datasets can be computed by

$$
\begin{aligned}
p_{i,j}^{sigmoid} &= sigmoid\Big(logit\_im_{i,j}\Big), && \text{if training datasets can have multi-span answers} \\
p_{i,j}^{softmax} &= softmax\Big(logit\_im\Big)_{i,j}, && \text{if training datasets only have single-span answers}
\end{aligned}
\tag{3}
$$

Target image, $target\_im$, is a binary image with zeros at every pixel except those corresponding to target spans. (*i.e.,* $target\_im(i, j) = 1$ for any target span in $g$ with start index $i$ and end index $j$). Given $logit\_im$ and $target\_im$, the loss function using sigmoid is given below

$$Loss(q, p, target\_im) = \sum_{i,j} H\Big(target\_im_{i,j}, p_{i,j}^{sigmoid}\Big)/N^2, \tag{4}$$

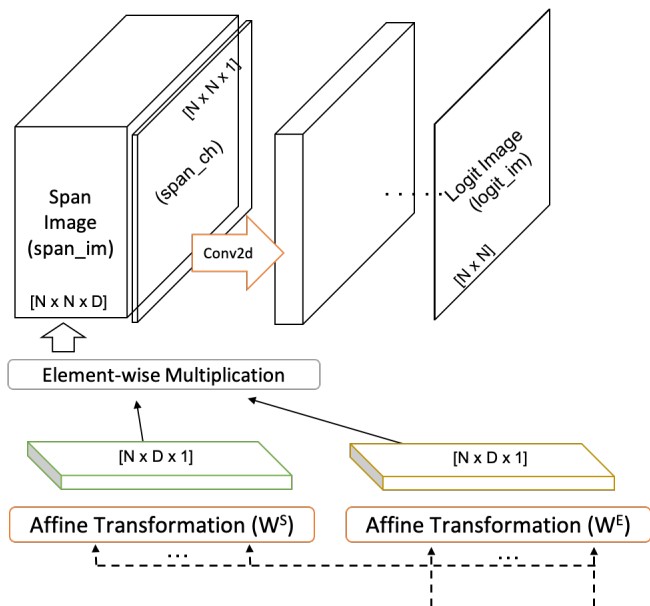

Figure 3: Span-image network implements convolution layers as typically used in image classification tasks.

where $H$ is the cross-entropy function. The loss function for softmax is given by

$$Loss(q, p, target\_im) = H\Big(target\_im, p^{softmax}\Big). \tag{5}$$

Note that $target\_im$ and $p^{softmax}$ are joint pdfs on $s^S$ and $s^E$, while $p_{i,j}^{sigmoid}$ is a pdf for binary variable indicating if $(i, j)$ is an answer span or not.

### 2.2.1 INCORPORATING SPAN CHARACTERISTICS

Typical post-processing on BERT QA output involves checking for valid spans and sorting them with the multiplication of start/end index probabilities. As shown in Figure 1, a priori information about the span length distribution can be used to break ties or prefer between spans that have close probabilities. In BERT QA, this can only be achieved by implementing a post-processing technique that penalizes spans based on their lengths. Our span-image architecture, however, has an inherent capability to learn and incorporate such patterns. We simply create a new channel $span\_ch$, and our model learns how to utilize this channel to capture span-characteristics during training.

$$span\_ch = \begin{cases} -1 & \text{if } i < j \\ j - i & \text{if } j \geq i, j - i < \varsigma \\ \varsigma & \text{if } j - i > \varsigma. \end{cases} \tag{6}$$

$span\_ch$ is concatenated to $span\_im$ increasing its depth to $D + 1$.

## 3 EXPERIMENTS

### 3.1 IMPLEMENTATION DETAILS

In multi-span QA tasks, we compare the standard BERT-QA model by Devlin et al. (2019), which has a separate output layer for start and end index prediction, against the following variants of the span-image architecture, which is described in Section 2.2:

- **bert-qa**: The BERT base model that is available from Transformers (Wolf et al., 2019) library as *bert-base-uncased*.

- **bert-ms-sigmoid**: Consists of the BERT base model augmented by the span-image network, which involves two 2D convolution layers with 100 and 50 filters, respectively. Both layers use 3x3 filters. The output layer has sigmoid activation on span-image pixels to enable multiple span predictions.

- **base-ms-softmax**: Replaces the sigmoid activation of the *bert-ms-sigmoid* with softmax activation. Softmax serves as a useful regularization when task dictates one answer (no-answer counts as a "null" answer).

- **bert-ms-sigmoid-sl**: *bert-ms-sigmoid* model with a span-length indicator channel concatenated to the span-image.

- **bert-ms-softmax-sl**: *bert-ms-softmax* model with a span-length indicator channel concatenated to the span-image.

## 3.2 SQUAD

The Stanford Question Answering Dataset (SQuAD v1.1) is a collection of 100,000+ crowd-sourced question/answer pairs (Rajpurkar et al., 2016). The SQuAD v2.0 task extends the SQuAD 1.1 problem with question/answer pairs, in which there may be no answer to the question. We use SQuAD v2.0 in all our experiments. This makes the QA task more realistic and challenging.

## 3.3 INTERNAL AMAZON MULTI-ANSWER DATASET

Consumers use the total quantity information of a product to compare its value offer against similar products. To provide Amazon customers with accurate quantity information, we formulated the quantity extraction as a multi-span question answering problem where the question is "*what is the total quantity of this item in terms of its unit of measure?*", and the context is the textual item description provided by sellers. The unit of measure can be volume (*e.g, liquid detergent*), weight (*e.g., powder detergent*), or count (*e.g., number of loads*) depending on the product type. To compute the total quantity correctly, all relevant quantities need to be extracted from the seller-provided text and multiplied. For example, if the product title is "*Original Roast Ground Coffee K Cups, Caffeinated, 36 ct - 12.4 oz Box, Pack of 2*", then the applicable question is "*what is the count?*", and the total quantity is "*36 x 2 = 72*". Our dataset consists of manual labels accumulated in time. We split 80,000 out of 450,000 labels into a test set. The average number of answer spans in our dataset is 1.8.

## 3.4 RESULTS

The performance metrics on fine-tuning BERT with SQuAD v2.0 dataset are given in Table 1. All models perform similarly with the exception of *bert-ms-sigmoid*. This is expected since SQuAD 2.0 dataset has only one answer, the softmax activation exploits this information by its inherent normalization, which can be seen as a projection to the single-answer constraint.

Table 1: SQuAD v2.0 results for top answer prediction

| Model Name | EM | F1 | total | HasAns-EM | HasAns-F1 | HasAns-total | NoAns-EMt | NoAns-F1 | NoAns-total |
|---|---|---|---|---|---|---|---|---|---|
| bert-qa | 63.6 | 67.2 | 11873 | 63.2 | **70.2** | 5928 | 64.1 | 64.1 | 5945 |
| bert-ms-softmax | 64.0 | 66.8 | 11873 | **63.3** | 68.8 | 5928 | 64.7 | 64.7 | 5945 |
| bert-ms-softmax-sl | **64.9** | **67.4** | 11873 | 62.5 | 67.5 | 5928 | **67.3** | **67.3** | 5945 |
| bert-ms-sigmoid | 59.7 | 62.3 | 11873 | 59.4 | 64.6 | 5928 | 60.0 | 60.0 | 5945 |

Since span-image architecture enables BERT to generate multi-span answers, we use top-K accuracy to compare *bert-qa* with *bert-ms-softmax-sl* for K=1,3,5, and 10. As shown in Table 2, both models perform similarly for K=1, but *bert-ms-softmax-sl* model significantly performs better in all metrics for $K > 1$.

### 3.4.1 AMAZON INTERNAL MULTI-SPAN ANSWER DATASET

Using top-K accuracy is a proxy to capture answer quality on SQuAD when multiple answers are output by the models. Our internal quantity dataset is a true multi-span answer dataset that we

Table 2: SQuAD results using top-K answers

| Row Labels | EM | F1 | HasAns-EM | HasAns-F1 | NoAns-EM | NoAns-F1 |
|---|---|---|---|---|---|---|
| **K=1** | | | | | | |
| bert-qa | 63.24±0.23 | **66.71±0.25** | **63.60±0.25** | **70.54±0.20** | 62.89±0.7 | 62.89±0.7 |
| bert-ms-softmax-sl | **64.12±0.37** | **66.71±0.33** | 62.90±0.48 | 68.07±0.48 | **65.35±0.49** | **65.35±0.49** |
| **K=3** | | | | | | |
| bert-qa | 80.44±0.14 | 83.08±0.18 | 83.05±0.01 | 88.34±0.09 | 77.83±0.27 | 77.83±0.27 |
| bert-ms-softmax-sl | **88.42±0.28** | **90.53±0.19** | **86.04±0.49** | **90.25±0.27** | **90.80±0.39** | **90.80±0.39** |
| **K=5** | | | | | | |
| bert-qa | 85.39±0.18 | 87.53±0.18 | 87.23±0.08 | 91.51±0.07 | 83.56±0.29 | 83.56±0.29 |
| bert-ms-softmax-sl | **93.44±0.16** | **94.96±0.15** | **90.74±0.14** | **93.79±0.08** | **96.13±0.24** | **96.13±0.24** |
| **K=10** | | | | | | |
| bert-qa | 90.45±0.11 | 92.09±0.07 | 90.93±0.14 | 94.23±0.06 | 89.96±0.08 | 89.96±0.08 |
| bert-ms-softmax-sl | **96.81±0.10** | **97.83±0.06** | **94.63±0.18** | **96.66±0.12** | **98.98±0.09** | **98.98±0.09** |

fine-tune BERT with. We measure performances on multi-span answer prediction (*i.e., whether all relevant information to compute the total quantity has been extracted or not*). On this task, since the span count can be any number, we compare *bert-ms-sigmoid* and *bert-ms-sigmoid-sl* with *bert-qa*. While both of the span-image models perform better, *bert-ms-sigmoid-sl* performs best. Making the QA model aware of span-length leads to small improvements.

Table 3: Question-answer pairs in the multi-answer dataset

| Paragraph | Question | Answers |
|---|---|---|
| Blackstrap Molasses Yummmy **5** Lbs, Kosher Certified, BPA free container, All Natural, Unsulfured Sale by weight | What is the weight? | [5] |
| Sky Organics Grapeseed Oil 100% Pure, Natural & Cold-Pressed - Ideal for Massage, Cooking and Aromatherapy- Rich in Vitamin A, E and K- Helps Reduce Wrinkles, **8** oz (Pack of **2**) (Packaging may vary) | What is the volume? | [8, 2] |
| XUAN YUAN Fabric Shaver-Wool Clothes Pilling Trim Rechargeable Shaving Hair Removal Artifact Cleaner Tick Suction Household Ball Machine Fabric razor (Size : Spare cutter head x1) | How many are there? | [] |

Table 4: Multi-Answer Dataset Results

| Model Name | EM | F1 | total | HasAns-EM | HasAns-F1 | HasAns-total | NoAns-EM | NoAns-F1 | NoAns-total |
|---|---|---|---|---|---|---|---|---|---|
| bert-qa | 79.2 | 85.0 | 35728 | 64.4 | 76.3 | 17510 | **93.5** | **93.5** | 18218 |
| bert-ms-sigmoid | 88.9 | 90.3 | 35728 | 84.9 | **87.6** | 17510 | 92.8 | 92.8 | 18218 |
| bert-ms-sigmoid-sl | **89.1** | **90.4** | 35728 | **85.0** | **87.6** | 17510 | 93.0 | 93.0 | 18218 |

To improve *bert-qa*'s multi-span prediction performance, we introduce a post-processing step where a weighted penalty term for the span length is added to start/end index probabilities. Results for different span-length penalty weights ($\lambda$'s) are given in Table 5. Span-length penalization improves the performance of *bert-qa*, but it still performs worse than span-image models when an answer is present in the input text.

## 4 CONCLUSION

QA models can answer complex questions by predicting a span in a given paragraph or context. To associate a question and a context, large language-representation models, which can be trained on big corpora, are utilized. As the performance of QA models improves, the need for more realistic scenarios grows. In this work, we propose *span-image* network to predict multiple spans of an answer. We measure its performance using top-K accuracy on SQuAD and exact match of all spans on an internal Amazon multi-span QA dataset. While performing similarly on top 1 accuracy, span-image network significantly outperforms separable prediction for $K > 1$.

Table 5: Multi-answer dataset results using penalty weight $\lambda$

| Model Name | $\lambda$ | EM | F1 | total | HasAns-EM | HasAns-F1 | HasAns-total | NoAns-EM | NoAns-F1 | NoAns-total |
|---|---|---|---|---|---|---|---|---|---|---|
| bert-base-uncased | 0 | 79.2 | 85.0 | 35728 | 64.4 | 76.3 | 17510 | 93.5 | 93.5 | 18218 |
| bert-base-uncased | 5 | 81.9 | 86.8 | 35728 | 69.8 | 79.9 | 17510 | 93.5 | 93.5 | 18218 |
| bert-base-uncased | 10 | 87.3 | 89.7 | 35728 | 80.9 | 85.8 | 17510 | 93.5 | 93.5 | 18218 |
| bert-base-uncased | 20 | 88.5 | 90.1 | 35728 | 83.3 | 86.5 | 17510 | 93.5 | 93.5 | 18218 |
| bert-base-uncased | 25 | 87.5 | 89.2 | 35728 | 81.2 | 84.7 | 17510 | 93.5 | 93.5 | 18218 |
| bert-base-uncased | 30 | 86.8 | 88.6 | 35728 | 79.8 | 83.4 | 17510 | 93.5 | 93.5 | 18218 |

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
