# OpenReview forum: "MULTI-SPAN QUESTION ANSWERING USING SPAN-IMAGE NETWORK"
_ICLR.cc/2021/Conference — Reject_

### Official Review · AnonReviewer3 · 2020-10-22
**Blind Review**

**Rating:** 5
**Confidence:** 3

**Review:**

#### Summary
The paper proposes a novel method for predicting multiple answer spans in question-answering (QA) tasks. When the Span-Image technique is applied to a base BERT model, the authors show performance gains on a single-span dataset (SQuAD) and substantial improvements on a multi-span dataset (an internal Amazon dataset). The authors propose that the method is both model-agnostic and can eliminate common post-processing steps via built-in architecture design choices.
#### Positives
- The authors show significant improvement in top-K accuracy on the SQuAD dataset, highlighting their technique’s ability to hone in on multiple relevant spans.
- The authors show that the method works (either does not degrade performance or indeed improves performance) for both single-span and multi-span datasets.
- The technique is not too complex and is model agnostic (does not require BERT models to function)
- The paper is well-written, with clear descriptions of baseline architectures and their new Span-Image architecture and appropriate accompanying diagrams and results tables.

#### Negatives
- The authors only compare to one baseline, BERT-QA, and do not fully motivate how it is state-of-the-art on the respective datasets considered; a few other observations call into question the generalizability of the method:
    - The authors claim that the technique is modular and can be applied to other architectures, however only one architecture is considered.
    - The authors also claim that the method is amenable to specifically designed channels to eliminate post-processing, however only one such channel is considered.
- A central claim is that the method works better for multi-span question answering, however the authors only consider one true multi-span dataset (an internal dataset). There are other multi-span “reading comprehension” datasets in the literature (DROP[1], Quoref[2]) that would probably have been appropriate datasets to which the authors could have applied their technique.
- The significant performance gains of the authors’ technique on the multi-span dataset are mitigated almost entirely by adding an additional length penalty to the base BERT-QA model, raising the question about whether the authors’ technique itself is what effects better performance (rather than, say, the explicit length penalty channel in the architecture)

#### Decision
I think the paper is marginally below the acceptance threshold. The method is novel and explainable, and appears to improve performance on the two datasets over the BERT QA baseline. However, the method is not compared to other QA architectures (is BERT QA SOTA on SQuAD?), the gains do not appear to be entirely significant (especially on the multi-span dataset, where the performance gap is reduced significantly by adding a length penalty to the baseline model), and various proposed contributions are not expanded upon (e.g. the modularity of the architecture is not demonstrated). Ultimately, the paper could be improved by further exploration of the method, whether via application to new datasets or more in depth analysis of the generalizability of the method.

#### Questions
Mostly curious about the author’s responses to the questions posited above, and in addition:
1. Did the authors consider the RC datasets specified above (DROP, Quoref), or is there a reason they were omitted.
2. Did the authors try applying their method to other DNN architectures?
3. Did the authors consider other image channels to eliminate post-processing?
4. Did the authors try adding a specific span length penalty term to the Span-Image network as well?

#### Cites
[1] Dua, M. (2019). DROP: A Reading Comprehension Benchmark Requiring Discrete Reasoning Over Paragraphs. In Proceedings of the 2019 Conference of the North American Chapter of the Association for Computational Linguistics: Human Language Technologies, Volume 1 (Long and Short Papers) (pp. 2368–2378). Association for Computational Linguistics.
[2] Pradeep Dasigi, Nelson F. Liu, Ana Marasovic, Noah A. Smith, & Matt Gardner (2019). Quoref: A Reading Comprehension Dataset with Questions Requiring Coreferential Reasoning. In Proc. of EMNLP-IJCNLP.

---

### Official Review · AnonReviewer4 · 2020-10-27

**Rating:** 4
**Confidence:** 4

**Review:**

This paper proposes a new framework to inherently break the independence assumption of start-end decision making in extractive question answering models. The general idea is to expand the current point-to-point product to a 3-D convolution. Since each point absorbs information within the sliding window, the probability (i, j) does not solely depend on $h_i, h_j$. The basic motivation makes sense to me, but I still have the following concerns about the paper:
1) the proposed convolutional network though incorporates more information, it's constrained to be the nearest information in its sliding window, let's say 2 or 3, and the probability of p(i, j) thus depends on $h_{i-2:i+2}, h_{j-2:j+2}$. However, I don't think this modification will help the model make a better decision since the nearest surrounding words' information is already well encoded in the lower-layers of the transformer. The errors the model made are mainly due to a lack of global reasoning, the more further-away context is actually a critical issue for breaking the bottleneck. The proposed span-image framework doesn't seem to touch on the core problem in the current QA datasets.
2) the proposed method obtains similar or even worse performance on top-answer selection, this point kind of reflects my previous concern. Adding more neighboring word context does not help the model make a wiser decision, and thus the final scores remain similar to the standard BERT QA model.
3) the results in in-house datasets are not quite convincing. I think you can add some heuristics to the BERT QA answer span selection to improve its top-k results like adding diversity, prevent overlapping, etc. I'm not sure whether the gap will still remain as significant as reported in the paper.

---

### Official Review · AnonReviewer2 · 2020-10-28
**Not good enough for ICLR standard**

**Rating:** 1
**Confidence:** 5

**Review:**

This paper proposes a neural model called Span-Image Network that allows the model to output multiple spans as answers (existing BERT-based QA models usually only output two endpoints of the answer span). The model is evaluated on SQuAD (top-K answer prediction) and an internal Amazon dataset.

Unfortunately, it is clear that this paper doesn’t meet the standard of the ICLR publications. I can’t recommend the acceptance of the paper.

First of all, it is mentioned in the paper that “To the best of our knowledge, a multi-span QA architecture has not been proposed.” This is certainly incorrect. There have been many models proposed recently to tackle the multi-span QA problemm.  Some examples include:

- (Hu et al., EMNLP 2019): A Multi-type Multi-span Network for Reading Comprehension that Requires Discrete Reasoning
- (Segal et al, EMNLP 2020): A Simple and Effective Model for Answering Multi-span Questions
- (Andor et al., EMNLP 2019): Giving BERT a Calculator: Finding Operations and Arguments with Reading Comprehension => This one is not straightforward but they support merging single spans at the end

It is very natural to cast a multi-span QA problem as a sequence tagging task and this paper doesn’t have such a baseline to compare with.

Second, I think the model is not motivated well and the writing of the paper needs to be improved. Why is it called a span-image network? Is it because there is a CNN layer applied? What is the rationale behind that? It is really not an image of pixels. It is just an N by N matrix, and each entry is an elementwise-multiplication of the BERT hidden vectors.

Third, the evaluation of the paper also should be improved. SQuAD is a single-span QA dataset and using top-K prediction looks quite artificial and unnatural. Also, the performance on SQuAD 2 is also lower than expected. For a BERT-base-uncased model, it is expected to achieve at least ~75 F1 on SQuAD 2 so even the baselines don’t seem to be right. The internal Amazon dataset lacks details so I can’t comment much on that.  There are indeed several multi-span QA datasets (e.g., DROP, Quoref, Natural Questions) and I think the paper should experiment with those datasets and compare to previous approaches.

---

### Official Review · AnonReviewer1 · 2020-10-29
**Not ready for publication - missing important literatures, limited empirical justification, etc**

**Rating:** 3
**Confidence:** 4

**Review:**

This paper introduces a new QA model based on BERT, which is called Span-Image Network. The paper first points out that previous span extraction models model independent probability of the start and the end of the span, making the extension to multi-span extraction harder. Span-Image Network model the joint probability of the start and the end by deploying a 2-D convolution, enabling multi-span extraction.

I think the paper is not ready for publication for a few reasons.

First, the limitation of independent modeling of the start and the end of the span has been reported in the literature, and there have been a few works that use joint probability instead. I will only list some of them: work in QA [1] [2] as well as QA baselines for pretrained LMs [3][4][5]; [6], which is concurrent work to thsi work, includes detailed survey. Such literature has not been mentioned or discussed in this manuscript.

Second, although “multi” span extraction is the main motivation for this work, as described in the Introduction, there is no evaluation on public dataset. Experiment on Amazon internal data is included, however, as the detailed description or the data statistic is missing, it cannot be considered as academic empirical evaluation. The only experiment on public dataset is SQuAD which does not require multi-answer extraction and the proposed model does not show superior results compared to the baseline. (The top-k experiment in the paper is not convincing - it is synthetic and does not align with the goal of predicting multiple answers.) If the authors want to include public datasets for multi-span extraction, they can consider multi-answer portions of Natural Questions dataset [7] or AmbigQA dataset [8].

Third, as mentioned above, calculating alignment of the start and end position has been already incorporated in the previous work (such as [2]). The convolutional component is the one I have not seen in the previous literature - however, as there is no ablation with and without convolution, its effect is not shown in the paper.

Fourth, even with Amazon internal dataset, the gains are not significant compared to bert baseline with postprocessing. Based on Table 5, the baseline’s best EM is 88.5, whereas the best performance of the proposed model in Table 4 is 89.1. In fact, the baseline number in Table 4 is not the best number of the baseline, which will mislead the audience to believe that the gap is significant - another major issue of this paper.

Lastly, this is a minor point, but I believe “Span-Image” in the name of the model is largely misleading. There is no “image” involved in the model architecture or training. Something like “2D” or “convolution” might be a better term.


[1] Lee et al. Learning recurrent span extractions for extractive question answering. 2016.
[2] Seo et al. Real-time open-domain question answering with dense-sparse phrase index. 2019.
[3] Yang et al. Xlnet: Generalized autoregressive pretraining for language understanding. 2019.
[4] Lan et al. Albert: a lite bert for self-supervised learning of language representation. 2019.
[5] Clark et al. Electra: pre-training text encoders as discriminators rather than generators. 2020.
[6] Fajcik et al. Rethinking the objectives of extractive question answering. 2020.
[7] Kwiatkowski et al. Natural questions: a benchmark for question answering research. 2019.
[8] Min et al. Ambigqa: answering ambiguous open-domain questions. 2020.

---

### Decision · Program_Chairs · 2021-01-07
**Final Decision**

**Decision:**

Reject

**Comment:**

This paper is not ready for a publication at ICLR, as agreed unanimously by the reviewers.

There are three main reasons for that:
1. Novelty: it is mentioned in the paper that “To the best of our knowledge, a multi-span QA architecture has not been proposed", which is certainly incorrect. See the multiple references provided by the reviewers.
2. Evaluation: there is no evaluation in multi-span setting on a public dataset. SQuAD being single span, As stated by R1, "Experiment on Amazon internal data is included, however, as the detailed description or the data statistic is missing, it cannot be considered as academic empirical evaluation." Several public datasets could be used like DROP, Quoref, or Natural Questions.
3. Motivation: the reviewers also note that the clarity and motivation behind the work could be improved. Some choices of the architecture or the model should be more clearly justified.

We encourage the authors to look into the multiple comments in the reviews in order to improve the paper and the research project overall.